# Determination of the Roles of *H. pylori* Outer Membrane Virulence Factors and Pyroptosis-Associated NLRP3, ASC, Caspase-1, Gasdermin D, IL-1β, and IL-18 in Ulcer and Gastritis Pathogenesis

**DOI:** 10.3390/biology14060634

**Published:** 2025-05-30

**Authors:** Yaren Buyukcolak-Cebeci, Emel Timucin, Sumeyye Akcelik-Deveci, Nesteren Mansur-Ozen, Tuana Aydinlar, Arzu Tiftikci, Sinem Oktem-Okullu

**Affiliations:** 1Department of Medical Biotechnology, Institute of Health and Science, Acibadem Mehmet Ali Aydinlar University, Atasehir, Istanbul 34752, Turkey; yarenbyk@gmail.com (Y.B.-C.); smyy.akclk@gmail.com (S.A.-D.); nesterenmansur@gmail.com (N.M.-O.); 2Department of Biostatistics and Medical Informatics, School of Medicine, Acibadem Mehmet Ali Aydinlar University, Atasehir, Istanbul 34752, Turkey; emeldtimucin@gmail.com; 3Department of Molecular Biology and Genetics, Gebze Technical University, Gebze, Kocaeli 41400, Turkey; 4Department of Microbiology, Institute of Health and Science, Acibadem Mehmet Ali Aydinlar University, Atasehir, Istanbul 34752, Turkey; tuanaaydinlar@gmail.com; 5Department of Internal Medicine, School of Medicine, Acibadem Mehmet Ali Aydinlar University, Atasehir, Istanbul 34752, Turkey; arzu.tiftikci@acibadem.com; 6Department of Medical Microbiology, School of Medicine, Acibadem Mehmet Ali Aydinlar University, Atasehir, Istanbul 34752, Turkey

**Keywords:** *Helicobacter pylori*, outer membrane virulence factor, pyroptosis, programmed cell death, *H. pylori*-related gastritis, *H. pylori*-related ulcer

## Abstract

*Helicobacter pylori* is a prevalent gastric bacterium associated with gastritis and ulcers. This study explored how the bacterium may trigger pyroptosis—a form of inflammatory cell death—in stomach tissue. We analyzed tissue samples from patients with and without *H. pylori* infection to measure the activity of genes and proteins involved in this process. Our results showed that infection increased key pyroptosis markers, especially in ulcer cases. Certain bacterial genes, such as *vacA* and *babB*, were linked to stronger inflammatory responses, while another gene, *alpA*, was associated with reduced activation of a key pyroptosis protein. These findings suggest that *H. pylori* may influence disease severity by altering the host’s immune response.

## 1. Introduction

*Helicobacter pylori (H. pylori*) infection is seen in more than half of the world’s population and is the primary cause of acute and chronic gastritis, ulcers, and gastric cancer by colonizing the antrum and corpus of the human stomach. *H. pylori* is a spiral-structured, microaerophilic, gram-negative bacterium that moves via flagella, and the prevalence of *H. pylori* varies regionally and is lower in socioeconomically developed countries.

The specific properties of bacteria that are effective in reaching the gastric epithelium, surviving in the acidic environment, and adhering to the epithelial surface by avoiding the peristaltic movements of the stomach are called virulence factors. Virulence factors play an active role in the bacteria’s reaching the gastric epithelium, attachment, and initiation of colonization. While the urease activity of the bacteria supports their survival in the stomach’s acidic environment, the virulence factors CagA and VacA have been shown to play an active role in the development of pathogen-related diseases [1,2]. Bacterial outer membrane proteins interact with cellular receptors to protect the bacteria from mechanisms like acidic pH, mucus, and exfoliation of the stomach [3,4].

In the presence of AlpA, AlpB, and BabA proteins, an increased inflammatory response was observed [5,6]. and BabB and HopQ proteins were found to support the formation of gastric lesions [7,8]. In addition, the presence of BabA, SabA, and OipA proteins was evaluated as a biomarker for gastric cancer [9]. Although it is known that the outer membrane proteins of bacteria are very effective in the pathogenesis of the bacteria, it has not yet been clarified which mechanisms they affect in the development of diseases.

Pyroptosis is a lytic, inflammatory programmed cell death programmed against intracellular microbial pathogens; however, this beneficial mechanism can cause tissue damage by causing autoimmune and auto-inflammatory responses. NOD-like receptor family pyrin domain-containing 3 (NLRP3), Apoptosis-associated speck-like protein (ASC), Caspase-1, Gasdermin D (GSDMD), Interleukin-1 beta (IL-1β), and Interleukin-18 (IL-18) are the key molecules involved in pyroptosis. These molecules drive the pyroptotic process by leading to cell lysis and inflammation.

NLRP3 acts as a sensor, detecting cellular stress and damage, and is a key part of the inflammasome. ASC functions as an adaptor, linking NLRP3 to Caspase-1. Caspase-1 is an enzyme activated by the inflammasome, and it cleaves pro-inflammatory cytokines and GSDMD. GSDMD forms pores in the cell membrane, leading to pyroptosis. IL-1β and IL-18 are pro-inflammatory cytokines released during pyroptosis, amplifying the immune response. In the mechanism of pyroptosis, pathogen-associated molecular patterns (PAMPs) cause *NLRP3* inflammasome formation. Caspase-1, activated by this inflammasome, causes the differentiation of IL-1β and IL-18 from the pro-form to the active form, thus splitting the GSDMD protein into two. The GSDMD-N terminal integrates into the cell membrane and forms pores. Inflammatory cytokines are released from the pores on the cell surface into the surrounding environment, and osmotic lysis occurs due to fluid entry into the cell.

Moreover, it was observed that the expression of pyroptosis markers was higher in patients with gastric cancer infected with *H. pylori* [10]. In a study by Zhang, it was shown that *H. pylori* CagA virulence factor activates the NLRP3 inflammasome to promote gastric cancer cell migration and invasion. At the same time, in infection experiments using *cagA+* and *cagA*− *H. pylori* strains in 2021, the expression levels of NLRP3, ASC, IL-1β, and IL-18 were higher as a result of *cagA*+ *H. pylori* infection compared to *cagA*− strains. Similarly, infection with *cagA*+ strains has been shown to increase the NLRP3 inflammatory response and increase the potential for cancer and cell migration in the stomach by stimulating ROS generation [11]. Despite these, no studies have specifically explored the association between pyroptosis and *H. pylori* outer membrane proteins concerning pathogen-related diseases. To this end, this study seeks to address this gap in the existing literature.

## 2. Materials and Methods

### 2.1. Patient Distribution and Demographics

The number of samples to be included in the study was determined using statistical power analysis. The study included a total of 44 *H. pylori*-infected patients (22 gastritis, 22 ulcer), and 15 uninfected individuals (5 gastritis patients, 5 ulcer patients, and 5 patients with normal histology) were included in the study. In the a priori power analysis for single factor ANOVA, the type 1 error was taken as 0.05 and the type 2 error as 0.20. Based on studies of similar effects in the literature, a medium-sized effect was determined as f = 0.4. The analysis made for 3 independent groups calculated how many individuals should be taken per group for 80% power. The gender and age distribution of patients in each group is summarized in Table 1. Although no statistically significant differences were observed in gender or age distribution between groups (*p* > 0.05), these demographic characteristics were taken into account when interpreting gene expression patterns and correlation analyses.

The research permit required to continue the study was obtained from Acıbadem Mehmet Ali Aydınlar University and Acıbadem Healthcare Institutions Medical Research Ethics Committee (ATADEK). Gastric tissue samples to be included in the study were obtained from volunteer patients at Acıbadem Maslak Hospital, Gastroenterology Clinic in Istanbul, who underwent an endoscopy because of gastroduodenal diseases. Patients were chosen based on specific criteria for enrollment in the study. These criteria include being under 18 years old, over 65 years old with an active infection, having cancer or inflammatory disease, experiencing gastrointestinal bleeding in the past month, having previous gastrointestinal surgery, diagnosed with chronic liver failure or renal failure, being diabetic, pregnant, previously treated for *H. pylori* infection, receiving immunosuppressive therapy or steroids, using NSAIDs or antibiotics in the last three weeks, taking antisecretory medication in the last two weeks, or refusing to sign the informed consent form voluntarily. In tissue samples, *H. pylori* infection was determined by the pathology report.

### 2.2. Obtaining Patient Gastric Tissue Samples

Tissue samples taken from the antrum region of the stomach during gastroscopy were taken into 400 µL of stabilizer solution (Sigma-Aldrich, St. Louis, MO, USA) for stabilization of DNA, RNA, and protein in tissue extracts and stored at −80 °C. With the help of the unique composition of stabilizer solution, the isolation of proteins with functional activity while preserving the integrity of nucleic acids is available. *H. pylori*-negative individuals with gastritis and ulcers, and also *H. pylori*-negative healthy patients, were included in the study as a control group.

### 2.3. Isolation of DNA, RNA, and Total Protein from Patient Tissue Samples

*H. pylori*-infected 22 gastritis patients and 22 ulcer patients were selected from the collected tissues. As the control group, 5 gastritis and 5 ulcer patients, and 5 volunteers with normal histological features without bacterial infection were selected. Tissue samples were fragmented using a homogenizer (BioSpec, Bartlesville, OK, USA) with metal beads. DNA and RNA isolation were performed as specified in the protocol of the Duet DNA/RNA MiniPrep Plus kit (Zymo Research, Irvine, CA, USA), and the acetone precipitate was used for total protein isolation. During the protein isolation period, four times the volume of cold acetone was added to the lysate expressing the protein content and incubated at −20 °C for one hour after vortexing. After centrifugation at 13.000× *g* for 10 min, the supernatant was removed, and the protein-containing pellet was left to dry for 30 min under a hood to remove the remaining acetone. The resulting pellets were dissolved by vortexing in 250 µL, 160 mMTris-HCl (pH = 6.8) containing 2% SDS. The concentrations of the obtained DNAs and RNAs were measured in the nanodrop device, and the BCA protein assay kit (Pierce, Thermo Fisher Scientific, Rockford, IL, USA) was used to determine the concentration of protein samples.

### 2.4. Investigation of the Presence of Bacterial Virulence Genes by Conventional PCR

The presence of *ureA*, *ureB*, *cagA*, *vacA m* and *s* alleles, and *alpA*, *alpB*, *sabA*, *sabB*, *hopQ*, *hopZ*, *oipA*, *labA*, *babA2*, *babB*, and *babC* genes in *H. pylori*-infected patient groups was investigated by PCR reaction. The genomic DNA of the *H. pylori G27* strain, which is known to express all virulence genes of the bacteria, was used as a positive control in the study. All primers used in the study are given in Table 2 together with their annealing temperatures. After the PCR reaction, the samples were run in a 1.5% agarose gel and stained with EtBr, and the presence/absence of virulence genes was detected by the ChemiDoc (Biorad, Hercules, CA, USA) instrument.

### 2.5. Investigation of the Expression of NLRP3 and ASC Levels by Real-Time PCR

Using the obtained cDNA products, the expression levels of *NLRP3* and *ASC* in patient samples were investigated at the RNA level. *GAPDH* gene-specific primer was used as a housekeeping gene in the experiments, and normalization was done according to the GAPDH expression of patient samples. Primers used in RT-PCR experiments can be seen in Table 2.

### 2.6. Evaluation of Expressions of GSDMD, Caspase-1, IL-18, and IL-1β by Western Blot

To compare the expression levels of the markers known to be activated in the pyroptosis process, protein samples belonging to different patient groups were run on 12% SDS-PAGE and PVDF membrane by the wet transfer method. The membrane was blotted with GSDMD (1:750, St John’s Labs, London, UK), caspase-1 (1:1000, St John’s Labs, London, UK), IL-18 (1:1000, St John’s Labs, London, UK), and IL-1β (1:2000, St John’s Labs, London, UK), which can detect pro and active forms of target markers. After the target bands were detected with ChemiDoc (BioRad, Hercules, CA, USA), the membrane was stripped, bound antibodies were removed, and the membrane was reblotted with β-actin (1:1000, Cell Signaling) for normalization. For the stripping, the membrane was washed three times with strip buffer (1-L strip buffer components (pH = 2.2): 15 g glycine, 1 g SDS, 10 mL Tween-20) for 10 min. Afterward, the membrane was washed twice with PBS and once with TBS-T buffer for 10 min. As outlined in Appendix A, imaging was conducted using the ChemiDoc system, and band intensities were quantified with the ImageLab software (Version 5.0 build 18). Patients with comparable protein concentrations were grouped, ensuring representation from each control group.

### 2.7. Statistical Analysis

To examine the effects of bacterial virulence genes and pyroptosis marker expression on pathogen-related diseases, we analyzed the experimental data using GraphPad Prism (9.0.0). The distribution of virulence genes in gastritis and ulcer patients was evaluated using the Chi-square test, and associated odds ratios were calculated. The relationship between ASC, NLRP3, and both forms of GSDMD, Caspase-1, IL-18, and IL-1β expression levels was assessed using the Kruskal-Wallis test for each patient.

Based on qRT-PCR and Western blot results, a correlation matrix was generated for each patient group to examine the relationship between the marker expression levels. To investigate the frequency of co-expression, Pearson correlation coefficients and *p*-values were calculated. Additionally, scatter plots were used to explore the relationship between pro and active forms of target markers in each patient group. In patients with increased pyroptosis responses, *p*-values were calculated to assess the presence of bacterial virulence genes. For each virulence gene, patients expressing the gene were selected from the population, and the data were analyzed using the Kruskal-Wallis test. The significance value was set to 0.05 for all analyses.

## 3. Results

The distribution of infection severity levels among *H. pylori*-infected gastritis and ulcer patients is given in Figure 1. The majority of individuals with a severity level 1 (mild infection) were found to have gastritis. Patients with ulcers were more likely to have a severity level 2 infection. It’s interesting to note that individuals with gastritis were more likely to have the greatest infection severity (level 3). Yet based on the statistical analysis using the chi-square test showed no significant association between infection severity and clinical diagnosis between the two groups (*p* = 0.81). show the severity of gastric inflammation and ulceration based on histopathological scoring and include correlation analyses between severity scores and gene expression levels. Appendix A show specific information on the degree of stomach inflammation and ulceration as evaluated by histological grading. These data also provide correlation studies between severity scores and gene expression levels. The histopathological scores were used to measure the degree of inflammation and ulceration, while correlation analyses were conducted to investigate possible correlations between the severity of stomach damage and the expression of key genes implicated in the pathogenesis of *H. pylori* infection.

To visualize the distribution of patients according to their virulence gene profiles, pyroptosis marker expression, and infection severity, heat map analysis was performed. Results are presented in the Appendix A.

The presence of virulence genes of *H. pylori* in DNA samples isolated from patient tissues was investigated by conventional PCR. The frequency of virulence genes in patient groups and the *p*-values and odds ratios calculated using Pearson’s Chi-square and Fisher’s Exact test are given in Table 3. While the *ureB*, *hopZ*, *hopQ*, and *alpA* genes were seen in all ulcer patients, the *ureA* gene was found in all gastritis and ulcer patients infected with *H. pylori*. No variability was found in the prevalence of the *sabB* gene in patients with gastritis and ulcers. The *babA2*, *babB*, *babC*, and *labA* genes were more common in gastritis patients than in ulcer patients. *ureB*, *cagA*, *oipA*, *sabA*, *alpA*, *alpB*, *hopZ*, and *hopQ* genes were seen more frequently in ulcer patients than in gastritis patients. Alleles of the *vacA* gene are frequently seen in *s1* and *m2* alleles in ulcer patients, while the opposite is true for gastritis patients. Chi-square Fisher’s Exact test was performed using GraphPad Prism (9.0.0) to examine the distribution of virulence genes of bacteria in different patient groups.

The distribution of relative fold change ratios of the *ASC* and NRLP3 genes and both forms of GSDMD, caspase-1, IL-18, and IL-1β expression levels in different patient groups is given in Figure 2. The Kruskal-Wallis test was used to test the significance of the mean ratio of target markers to housekeeping markers expression in four different patient groups by using GraphPad Prism (9.0.0). The alpha value of 0.05 was chosen. It was found to be statistically significant when *p* < 0.05, and this is indicated by an asterisk. The graphs were prepared according to the mean with SEM value. Error bars in all graphs represent the mean ± standard error of the mean (SEM), unless otherwise indicated.

Error bars in all graphs represent the mean ± standard error of the mean (SEM), unless otherwise indicated. Sample sizes: 22 *H. pylori*-infected gastritis patients, 22 infected ulcer patients, and 15 controls (5 uninfected gastritis patients, 5 uninfected ulcer patients, and 5 healthy volunteers with normal histology). Statistical analysis was performed using one-way ANOVA followed by Tukey’s post hoc test; *p* < 0.05 was considered statistically significant.

Changes in *ASC* and *NLRP3* gene expressions were not significant for any patient group (Figure 2A). It has been observed that the *ASC* gene is expressed more in gastritis patients than in ulcer patients. In addition, *H. pylori* infection increased the expression levels of both *ASC* and *NLRP3* in ulcer patients.

When the expression of the pro and active forms of Caspase-1 was examined, *H. pylori* infection increased the expression in both gastritis and ulcer patients. A similar relationship was also seen for pro-GSDMD expression, and the pro-GSDMD variation between uninfected gastritis and infected gastritis expression levels was statistically significant. Expression of the pro and mature forms of IL-18 is decreased with *H. pylori* infection, especially the amount of pro-IL-18, which is significantly higher in uninfected gastritis patients than in infected patients. Mature IL-1β was upregulated in the presence of ulcers in both control and infected patients (Figure 2B). Patients were grouped based on comparable tissue protein concentrations, with careful inclusion of control group representatives in each group (Appendix A).

To investigate the relationship between mature and pro-forms of target markers, correlation matrices were created, and the linearity and strength of the relationship between pro and active forms were evaluated. Heatmaps were created using GraphPad Prism (9.0.0) (Figure 3).

In uninfected ulcer patients, a significant correlation was seen between *NLRP3* expression and IL-1β expression. Similarly, there is a relationship between the active forms of caspase-1 and IL-18. Besides, a negative correlation was observed between the expression levels of *NLRP3* and *ASC* genes. The correlation between expression levels of mature caspase-1 and mature IL-18 was significant in patients with gastritis infected with *H. pylori*. No relationship was found between other markers in this patient group. Similarly, active caspase-1 and GSDMD-N levels were found to be significantly correlated in *H. pylori*-infected ulcer patients (Figure 3).

In addition, the relationship between pro and active forms of target markers was examined; graphs showing the balance between active and pro forms of target markers in each patient sample are given in Figure 4. Graphs were created using GraphPad Prism (9.0.0).

In patients with gastritis (72.2%) and ulcers (84.6%) infected with *H. pylori*, concomitance between pro and active forms of caspase-1 is more common than in patients with gastritis (40%) and ulcers (50%) without infection. While the expression of pro and mature forms of caspase-1 is increased in 44.4% of infected gastritis patients compared to the control group, the prevalence of this situation in infected ulcer patients is 41.2%. In addition to all these, in 4 infected gastritis patients (22.2% of the population) and 6 infected ulcer patients (35.3% of the population), caspase-1 expression was lower in the pro-form compared to the control group, while the expression level was increased in the active form (Figure 4A,B).

The expression of the active and pro-forms of GSDMD was generally decreased compared to the control group, and the relationship between the pro and active forms was completely simultaneous in uninfected gastritis and ulcer patients (Figure 4A,B). In the infected patient groups, GSDMD expression changed simultaneously in 66.7% of the population in the presence of gastritis and synchronously in 80% of ulcer patients. Expression of both forms of GSDMD was decreased in 38.9% of infected gastritis patients, but this was seen in 70% of infected ulcer patients. Uninfected patients did not show highly mature GSDMD-N and less pro-GSDMD expression, as was the case in 27.8% of infected gastritis patients and 10% of infected ulcer patients.

IL-18 expression is frequently altered simultaneously in infected gastritis patients (60%). Relative fold change values were found to be less than 1 in 45% of this patient group. In patients with pathogen-infected ulcers, this ratio is 38.1%. Also, increased mature IL-18 expression and low pro-IL-18 levels were the most common in the infected ulcer population (42.9% of the population). Similarly, while mature IL-18 was upregulated, infected gastritis patients in whom the pro form was downregulated (20% of the population) were more common than when the two forms were upregulated together (Figure 4A,B).

Expression of IL-1β tended to be simultaneous when down-regulated in uninfected patient groups, but expression of IL-1β was not commonly found in infected patient groups. Simultaneity was observed in 66.7% of infected gastritis patients, whereas only 36.4% of infected ulcer patients were similarly identified. In *H. pylori*-infected ulcer patients, upregulation of active IL-1β and downregulation of pro-IL-1β were seen in 45.5% of the population, compared with 26.7% of infected gastritis patients (Figure 4A,B).

To examine the relationship between the increased expression of pyroptosis markers and the virulence factors of *H. pylori*, the presence of virulence factors in patient tissues where each marker was upregulated was investigated by GraphPad Prism (9.0.0), and odds ratios are given in Table 4. When the presence of the *alpA* gene was examined in patients with increased GSDMD expression, it was found to be statistically significant. Appendix A provide detailed graphical representations of the data related to the change in pyroptosis response in the presence of bacterial virulence genes in infected gastritis and ulcer patients.

The patient population was divided into two groups based on the presence of virulence genes. Kruskal-Wallis analysis was performed to compare the expression levels of pyroptosis markers across these groups. The significance level was set to 0.05. Statistically significant results were indicated by an asterisk (** *p* < 0.05). The graphs in the Appendix A show mean and SEM values. In gastritis patients, the presence of the *vacA m2* allele, the *ASC* gene was significantly downregulated compared to the uninfected gastritis patient group. When infected ulcer patients were examined, it was observed that the presence of *bab* and GSDMD-N expression was found to be significantly upregulated compared to their absence of these genes. In particular, the expression of the *ASC* gene was significantly increased compared to uninfected ulcer patients, and the presence of the *H. pylori alpA* gene was substantially correlated with the lack of Gasdermin D expression (OR = 0, *p* < 0.01), indicating that *alpA*-positive strains may inhibit pyroptosis-related pathways or escape host immune responses (Table 4).

## 4. Discussion

Identifying strain-specific virulence factors offers valuable insight into predicting disease outcomes and tailoring therapeutic strategies, particularly in cases involving highly virulent *H. pylori* strains. Among these factors, VacA contributes to mucosal injury through its vacuolating and cytotoxic effects, while outer membrane proteins (OMPs) facilitate bacterial adhesion and immune evasion. Although the involvement of OMPs in *H. pylori* infection is well-established, their role in modulating pyroptosis—a pro-inflammatory programmed cell death pathway—remains insufficiently explored.

This study combined molecular profiling of bacterial virulence determinants with analysis of pyroptosis markers in individual gastric biopsies to investigate the interplay between bacterial traits and host programmed cell death mechanisms. In our cohort, the prevalence of *H. pylori* infection (41.2%) was notably lower than historical reports from Turkey (>80%), likely reflecting the higher socioeconomic status of patients treated at a private hospital in Istanbul [20].

The distribution of *H. pylori* infection severity levels among patients diagnosed with gastritis and ulcers was explored, and findings showed that the majority of patients with mild infections (severity level 1) had gastritis, but patients with ulcers were more frequently associated with intermediate infections (severity level 2). Remarkably, individuals with gastritis also showed an increased risk of developing a serious infection (level 3). Nevertheless, statistical analysis employing the chi-square test revealed no significant correlation between the two groups’ clinical diagnoses (gastritis vs. ulcer) and the severity of the infection (*p* = 0.81). This finding implies that while there were variations in the two clinical groups’ levels of infection, these variations were not statistically significant. The lack of a substantial correlation between infection severity and clinical diagnosis might be attributed to several causes. First, it’s possible that the sample size was insufficient to identify minute variations in infection severity across groups. Variability in infection severity within each group may also be a result of the heterogeneity in the clinical development of *H. pylori* infection, which is driven by host variables, bacterial strain diversity, and environmental factors. The observed patterns might potentially have been impacted by the presence of other underlying diseases or other unaccounted-for clinical factors. Clarifying the connection between infection severity and the clinical diagnosis of gastritis vs. ulcers may require future research with bigger sample sizes and a more complete collection of clinical and microbiological parameters.

Analysis of virulence gene distribution revealed that *babA2*, *babC*, and *labA* were more frequently detected in gastritis patients, whereas *ureB*, *cagA*, *oipA*, *sabA*, *alpA*, *alpB*, *hopZ*, and *hopQ* were more prevalent in those with ulcers. The predominance of vacA s1 and m2 alleles among ulcer patients aligns with previous studies linking these alleles to more severe gastric pathology. Assessment of the odds ratios further supported the associations between specific genes, especially *cagA*, *vacA*, and *sabA*, and the development of ulcers.

Regarding pyroptosis markers, ASC expression was elevated in gastritis patients, and infection further increased ASC levels in ulcer patients compared to their uninfected counterparts, mirroring patterns observed in recent clinical studies [21,22]. *H. pylori* infection also resulted in upregulated NLRP3 and active caspase-1 expression in both gastritis and ulcer groups.

Interestingly, while pro-GSDMD and active GSDMD levels rose mainly in infected gastritis patients, the increase in active GSDMD did not reach statistical significance. Pro-IL-1β levels increased with infection in gastritis cases but declined in ulcer patients, whereas mature IL-1β was elevated across both infected groups. These observations are consistent with prior findings suggesting an early stimulation of IL-1β by *H. pylori* during initial infection stages. Conversely, IL-18 levels were reduced in infected gastritis cases, diverging from previous reports, although animal models suggest temporal variability in IL-18 responses, potentially explaining our results.

Correlation analyses revealed that ASC positively correlated with IL-18 and caspase-1 levels, whereas IL-1β negatively correlated with *ASC*, GSDMD-N, and IL-18. In uninfected ulcer patients, NLRP3 expression correlated with IL-1β, whereas in infected groups, active caspase-1 was closely associated with both GSDMD-N and mature IL-18, supporting canonical pyroptosis activation.

Further analysis of expression patterns indicated that caspase-1 pro-forms and active forms were often co-upregulated following infection. In contrast, for GSDMD, IL-18, and IL-1β, asynchronous patterns were frequently observed, suggesting activation of existing pools rather than de novo synthesis. Notably, mature forms of caspase-1, IL-18, and IL-1β were upregulated even when their precursor forms were suppressed, underscoring post-translational regulation characteristic of pyroptosis.

Linking bacterial virulence genes to pyroptotic activity, our data demonstrated higher frequencies of *cagA*, *babB*, and *vacA s1* alleles in patients displaying heightened pyroptosis markers. Conversely, *oipA*, *ureB*, *sabA*, and *vacA s2* alleles were inversely associated with pyroptosis, highlighting complex roles in host-pathogen interactions. Although OipA and SabA have been implicated in ulcerogenesis, their inverse relationship with pyroptosis markers observed here may be attributed to the functional “on/off” variability of these proteins, emphasizing the limitations of purely genotypic analyses. Moreover, BabA2 presence correlated with increased pyroptosis in gastritis but was reduced among ulcer patients, a finding warranting further investigation given BabA’s established role in severe gastric disease [23,24,25].

Kruskal-Wallis analysis showed that *hopQ* and *sabA* genes were linked to more frequent pyroptotic marker upregulation. While the overall presence of *sabA* decreased among pyroptotic patients, when expressed, it strongly promoted pyroptosis, suggesting a dual role depending on the context of the infection.

Additionally, GSDMD-N levels were significantly higher in infected ulcer patients harboring babB, and decreased ASC expression was noted in gastritis cases lacking *labA*, hinting at novel connections between underexplored virulence genes and pyroptosis pathways. Significant differences in ASC expression associated with *vacA m2* presence further support its contribution to inflammasome activation and ulcer formation [26,27,28,29].

Collectively, our findings identify *vacA m2* and *babB* as promising virulence factors involved in pyroptosis, offering new perspectives for research into *H. pylori*-associated pathology.

Our investigation found a strong link between the presence of the *H. pylori alpA* gene and the absence of Gasdermin D expression (OR = 0, *p* <0.01). Gasdermin D is a critical protein involved in the execution of pyroptosis, an infection-induced inflammatory cell death process associated with immune response regulation. The lack of Gasdermin D expression in *H. pylori*-infected patients carrying the alpA gene shows that alpA-positive strains have acquired strategies to limit or avoid pyroptosis-related processes. This conclusion is consistent with earlier studies demonstrating that *H. pylori* has evolved methods to circumvent host immune responses, increasing its survival within the stomach mucosa. The *alpA* gene, which produces a virulence protein with immune-modulatory capabilities, may play a role in suppressing inflammatory responses and so aiding bacterial survival. Pyroptosis suppression may allow *H. pylori* to avoid identification and clearance by immune cells, thereby contributing to persistent infection and the development of diseases such as gastritis and ulcers. The absence of Gasdermin D expression *in H. pylori alpA*-positive strains indicates that the interplay between bacterial virulence factors and host immunological pathways is more complicated than previously thought. *H. pylori’s* capacity to alter host cell death pathways, such as pyroptosis, may be a key strategy for evading immune surveillance and establishing long-term infection. More research into the molecular processes behind this link, such as the involvement of alpA in influencing inflammasome activity and immune cell apoptosis, is needed to better understand the broader implications for *H. pylori*-related disorders. Overall, our findings shed new light on *H. pylori’s* immune evasion methods, indicating that the presence of virulence genes such as alpA may be critical to its capacity to modify host cell death pathways and maintain chronic infection. Future research into the interactions between *H. pylori* virulence factors and host immune responses will be critical for designing tailored therapy techniques that disrupt these immune evasion mechanisms [30,31,32,33,34].

This study demonstrates that *H. pylori* infection modulates key pyroptosis markers, including ASC, NLRP3, caspase-1, GSDMD, IL-18, and IL-1β, and highlights significant associations between bacterial virulence factors and disease outcomes.

The identification of *vacA m2* and *babB* as potential drivers of pyroptosis opens new avenues for investigating *H. pylori*-induced gastric pathology. The notable correlation identified between the presence of the *H. pylori alpA* gene and the lack of Gasdermin D expression indicates a possible immune evasion strategy employed by *H. pylori* to inhibit host pyroptotic responses; this discovery not only enriches our comprehension of bacterial pathogenesis but also underscores *alpA* as a potential molecular target for forthcoming therapeutic approaches aimed at reinstating inflammasome activity and bolstering host defense. Despite inherent limitations, such as the relatively small sample size and tissue sampling restricted to the antrum, this study provides valuable insights into how bacterial virulence and host cell death mechanisms intersect in gastric disease. It should be noted that some visual differences observed between groups did not reach statistical significance, potentially due to high intra-group variability. This variability may have reduced the statistical power to detect certain differences. Future studies with larger sample sizes and reduced variability would provide clearer insights into these associations. Expanding research into more diverse and larger cohorts, including gastric cancer patients, will be critical for fully elucidating these mechanisms and their clinical relevance.

## 5. Conclusions

This work underscores the crucial influence of *H. pylori* infection on the modulation of pyroptosis markers, including ASC, NLRP3, caspase-1, GSDMD, IL-18, and IL-1β, while revealing substantial connections between bacterial virulence factors and disease development. The identification of vacA m2 and babB as potential drivers of pyroptosis offers new perspectives for understanding *H. pylori*-induced gastric pathology. Additionally, the association between the *H. pylori alpA* gene and the suppression of Gasdermin D expression suggests a possible immune evasion strategy, providing a promising molecular target for future therapeutic interventions aimed at restoring inflammasome function and enhancing host immune defense. While the study presents valuable insights, its limitations, including a small sample size and restricted tissue sampling, warrant consideration. These factors, along with high intra-group variability, the identification of *vacA m2* and *babB* as potential drivers of pyroptosis, offer new perspectives for understanding *H. pylori*-induced gastric pathology. Additionally, the association between the *H. pylori alpA* gene and the suppression of Gasdermin D expression suggests a possible immune evasion strategy, providing a promising molecular target for future therapeutic interventions aimed at restoring inflammasome function and enhancing host immune defense. While the study presents valuable insights, its limitations, including a small sample size and restricted tissue sampling, warrant consideration.

## Figures and Tables

**Figure 1 biology-14-00634-f001:**
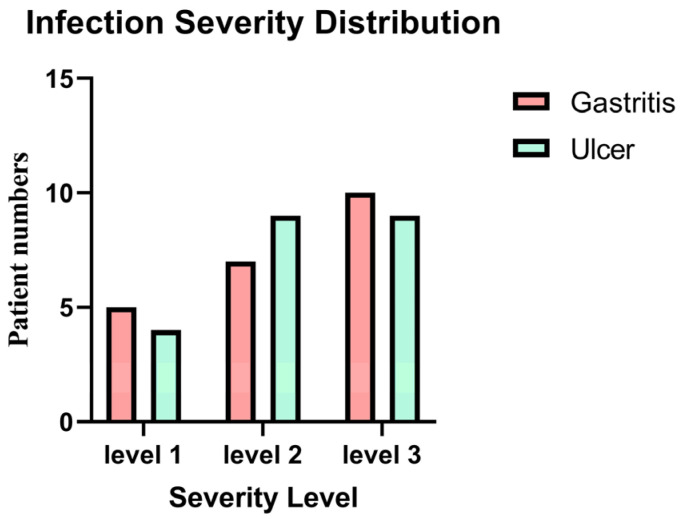
Distribution of infection severity levels among *H. pylori*-infected gastritis and ulcer patients. The figure shows the distribution of patients across three infection severity levels (1: mild, 2: moderate, 3: severe) in the gastritis and ulcer groups. Severity level 1 was more common in gastritis patients, level 2 in ulcer patients, and level 3 again in gastritis patients. However, no statistically significant association was found between infection severity and clinical diagnosis (*p* = 0.81, chi-square test).

**Figure 2 biology-14-00634-f002:**
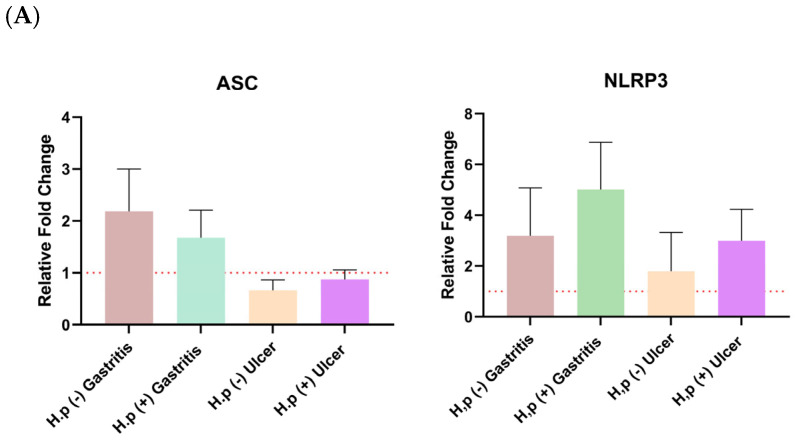
Distribution of relative fold change ratios in gene expression levels compared to the control group. (**A**) Relative mRNA expression levels of *ASC* and *NLRP3* genes. (**B**) Relative mRNA expression levels of both the full-length and cleaved forms of GSDMD, as well as caspase-1, IL18, and IL-1β in different patient groups (*H. pylori*-infected gastritis, infected ulcer, and uninfected controls). Gene expression was normalized to *GAPDH* and calculated using the 2^−ΔΔCt^ method. Dashed lines indicate thresholds for up-regulation (above the dashed line) and down-regulation (below the dashed line) in gene expression relative to the control group. The symbol * represent *p* < 0.05.

**Figure 3 biology-14-00634-f003:**
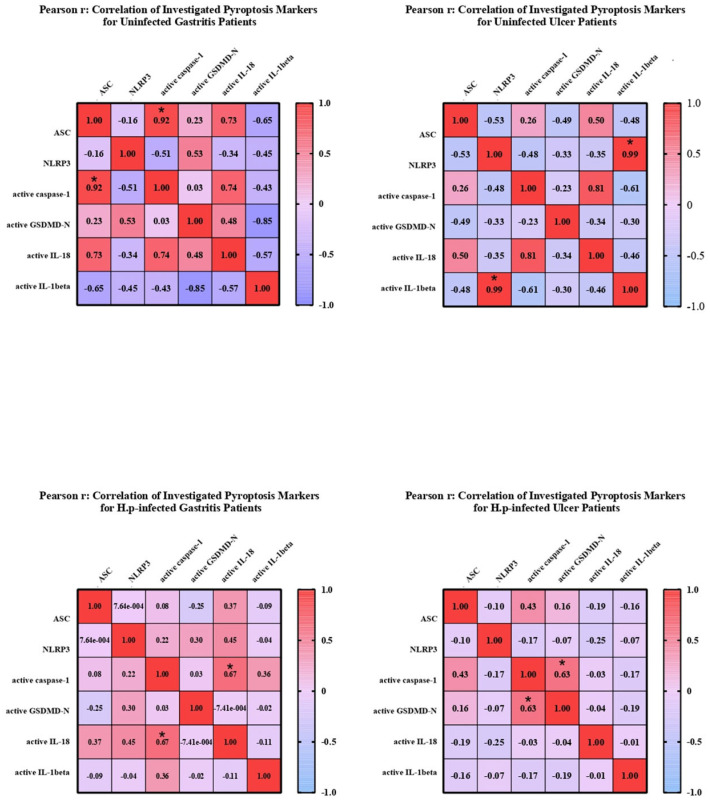
Correlation matrices showing the relationships between *ASC*, *NLRP3*, GSDMD, caspase-1−1, IL-18, and IL-1β expression levels. Pearson correlation coefficients (R) are displayed, with values approaching +1 (shown in red) indicating a strong positive relationship, and values approaching −1 (shown in blue) indicating a strong negative relationship between gene expression levels. Statistically significant correlations between genes are indicated by an asterisk (*p* <0.05). The analysis was performed using Pearson’s correlation, and data are shown for the following groups: 22 *H. pylori*-infected gastritis patients, 22 infected ulcer patients, and 15 controls (5 uninfected gastritis patients, 5 uninfected ulcer patients, and 5 healthy volunteers with normal histology). Statistical significance was determined using the Pearson correlation test with a significance level of *p* < 0.05. The symbol * represent *p* < 0.05.

**Figure 4 biology-14-00634-f004:**
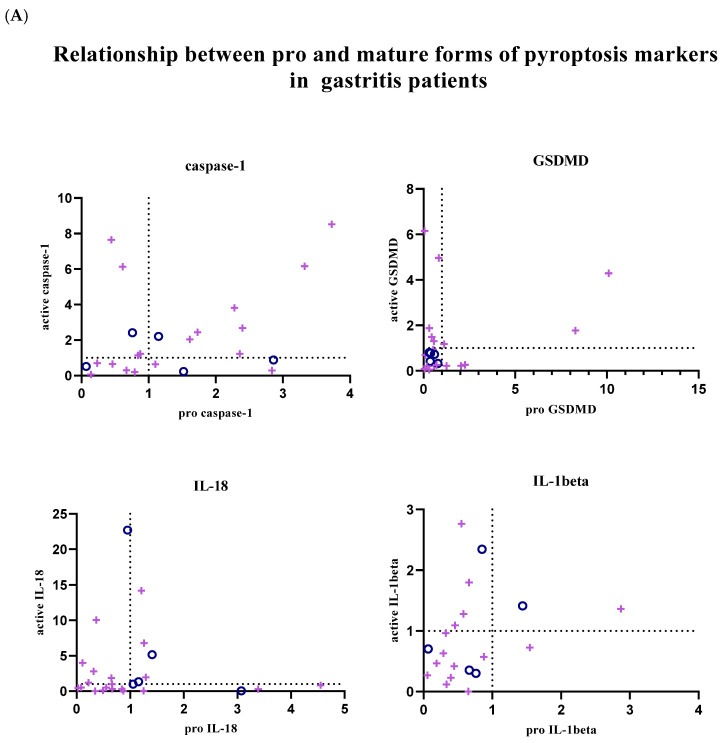
Graphical representation of individual gene expression data for each patient, comparing *H. pylori*-infected and uninfected cases. Each “+” symbol represents a *H. pylori*-infected patient, while each “o” symbol represents an uninfected patient. Fold change values and expression levels of the pro and active forms of the target markers (*GSDMD*, *caspase-1*, *IL-18*, and *IL-1β*) are shown. (**A**) Gastritis patients (n = 22 infected, 5 uninfected). (**B**) Ulcer patients (n = 22 infected, 5 uninfected). Gene expression was calculated using the 2^−ΔΔCt^ method and normalized to *GAPDH*. Data are plotted as individual values to highlight variation within each patient group. Statistical significance was assessed using unpaired t-tests or one-way ANOVA with post hoc analysis, as appropriate. A *p*-value < 0.05 was considered statistically significant.

**Table 1 biology-14-00634-t001:** The gender and age distribution of patients in each group.

		*H. pylori*-Infected Patients		Uninfected Patients		
		Gastritis	Ulcer	Gastritis	Ulcer	Control
**Patient Number**		22	22	5	5	5
**Gender**	Female	10	9	3	4	2
	Male	12	13	2	1	3
**Age**	18–30	1	3	-	-	1
	30–40	4	3	2	3	3
	40–50	8	5	2	1	-
	50–60	5	5	1	1	1
	>60	4	6	-	-	-

**Table 2 biology-14-00634-t002:** Primers are used in conventional PCR and real-time PCR experiments.

Primer	Sequence	Base Pair	Annealing Temperatures	References
***ureA*-F**	TGATGGGACCAACTCGTAACCGT	244 bp	60 °C	[12]
***ureA*-R**	CGCAATGTCTAAGCGTTTGCCGAA			
***ureB*-F**	AGTAGCCCGGTGAACACAACATCCT	645 bp	60 °C	[12]
***ureB*-R**	ATGCCTTTGTCATAAGCCGCTTGG			
***cagA*-F**	TTGACCAACAACCACAAACCGAAG	183 bp	58 °C	[13]
***cagA*-R**	CTTCCCTTAATTGCGAGATTCC			
***vacAs1s2*-F**	ATGGAAATACAACAAACACAC	259/286 bp	58 °C	[14]
***vacAs1s2*-R**	CTGCTTGAATGCGCCAAAC			
***vacAm1m2*-F**	CAATCTGTCCAATCAAGCGAG	570/645 bp	58 °C	[15]
***vacAm1m2*-R**	GCGTCAAAATAATTCCAAGG			
***alpB*-F**	TGCGACTGGTTCAGATGGTC	708 bp	58 °C	This study
***alpB*-R**	CTGAGCGTGGATTGGAAGGT			
***alpA*-F**	GGCTTACGCTACTACGGCTT	214 bp	58 °C	This study
***alpA*-R**	GGCTGTTTCTTAGCGTGCTG			
***sabA*-F**	TCGTCATCAGTGGCGTTTCA	621 bp	58 °C	This study
***sabA*-R**	GGTAGTTGGATTGGCCTGCT			
***sabB*-F**	GCTATCAAATCGGCGAAGCG	234 bp	58 °C	This study
***sabB*-R**	CTTGCGCGGTGTTGTAGATG			
***babA2*-F**	AATCCAAAAAGGAGAAAAAGTATGAAA	832 bp	55.5 °C	[16]
***babA2*-R**	TGTTAGTGATTTCGGTGTAGGACA			
***babB*-F**	GGTGGGCCTATATCCACTGC	803 bp	58 °C	This study
***babB*-R**	TGAGTGCCAAAGTGAGCGAT			
***babC*-F**	AACGGCGGTGTGTATCAGTT	755 bp	58 °C	This study
***babC*-R**	TGAGTGCCAAAGTGAGCGAT			
***labA*-F**	GCAGCGTTCGTGAAAGACTC	294 bp	58 °C	This study
***labA*-R**	CGCATCAGGCAAGCTAGAGA			
***hopQ*-F**	ACTCGGCTTCTGATGTGTGG	317 bp	58 °C	This study
***hopQ*-R**	TTTCACGCCCAATTCCATGC			
***hopZ*-F**	AACGGTGCGATGAATGGGAT	436 bp	58 °C	This study
***hopZ*-R**	TCTTCACGCCTAGTTCCACG			
***oipA*-F**	GTTTTTGATGCATGGGATTT	401 bp	60 °C	[17]
***oipA*-R**	GTGCATCTCTTATGGCTTT			
***GAPDH*-F**	CTCATGACCACAGTCCATGC	129 bp	58 °C	[18]
***GAPDH*-R**	TTCAGCTCTGGGATGACCTT			
***NLRP3*-F**	CAACCTCACGTCACACTGCT	170 bp	58 °C	[19]
***NLRP3*-R**	TTTCAGACAACCCCAGGTTC			
***ASC*-F**	CTGACGGATGAGCAGTACCA	224 bp	58 °C	[19]
***ASC*-R**	CAGGATGATTTGGTGGGATT			

**Table 3 biology-14-00634-t003:** Odds ratios and *p*-values of bacterial virulence genes in infected gastritis and ulcer patients.

Genes	Situation	Prevalence of Gastritis		Prevalence of Ulcer		Odds Ratio	*p*-Value
		n	(%)	n	(%)		
** *ureA* **	Present	22	100	22	100	-	>0.99
	Absent	0	0.0	0	0.0		
** *ureB* **	Present	20	90.9	22	100	2.1	0.49
	Absent	2	9.1	0	0.0		
** *cagA* **	Present	10	45.5	13	59.1	1.3	0.55
	Absent	12	54.5	9	40.9		
** *vacA* **	s1	17	77.3	19	86.4	-	-
	s2	3	13.6	3	13.6		
	Absent	2	9.1	0	0.0		
** *vacA* **	m1	10	45.5	10	45.5	-	-
	m2	11	50.0	12	54.5		
	Absent	1	4.5	0	0.0		
** *oipA* **	Present	15	68.2	18	81.8	1.4	0.3
	Absent	7	31.8	4	18.2		
** *babA2* **	Present	4	18.2	2	9.1	0.7	0.66
	Absent	18	81.8	20	90.9		
** *babB* **	Present	8	36.4	8	36.4	1	>>0.99
	Absent	14	63.6	14	63.6		
** *babC* **	Present	15	68.2	14	63.6	0.9	>>0.99
	Absent	7	31.8	8	36.4		
** *sabA* **	Present	20	90.9	21	95.5	1.4	>>0.99
	Absent	2	9.1	1	4.5		
** *sabB* **	Present	20	90.9	20	90.9	1	>>0.99
	Absent	2	9.1	2	9.1		
** *alpA* **	Present	21	95.5	22	100	2.1	>>0.99
	Absent	1	4.5	0	0.0		
** *alpB* **	Present	19	86.4	21	95.5	1.6	0.61
	Absent	3	13.6	1	4.5		
** *hopZ* **	Present	20	90.9	22	100	2.1	>>0.99
	Absent	2	9.1	0	0.0		
** *hopQ* **	Present	21	95.5	22	100	2.1	>>0.99
	Absent	1	4.5	0	0.0		
** *labA* **	Present	11	50.0	7	31.8	0.69	0.36
	Absent	11	50.0	15	68.2		

**Table 4 biology-14-00634-t004:** The relationship between the increased expression of pyroptosis markers and the virulence factors of *H. pylori*, with their odds ratios. ** indicates *p*< 0.01. NA: Not applicable.

In the Patient Population with Increased Target Pyroptosis Markers Based on the Odds Ratios						
	ASC	NLRP3	Active Caspase-1	Active GSDMD-N	Active IL-18	Active IL-1β
** *ureA* **	NA	0	NA	NA	3.2	0.33
** *ureB* **	NA	NA	NA	NA	NA	NA
** *cagA* **	1.47	1.71	0.42	1.86	1.2	0.48
** *vacA m1* **	1.75	2	0.24	NA	0	0
** *vacA m2* **	0.25	0.8	0.23	NA	0	0
** *vacA s1* **	NA	0	0	0	0	0
** *vacA s2* **	NA	NA	0	0	0	0
** *sabA* **	NA	1.46	0.42	NA	NA	0
** *sabB* **	0	1.46	1.36	NA	0.64	0
** *babA* **	0	0	0.19	0.73	0.3	0.25
** *babB* **	0.8	2	0.84	3.2	0.34	1.43
** *babC* **	1.3	6.86	0.73	0.75	0.72	0.20
** *hopQ* **	NA	NA	NA	NA	0.64	0
** *hopZ* **	NA	NA	NA	NA	NA	0
** *alpA* **	NA	NA	NA	0 **	NA	0
** *alpB* **	NA	NA	4.3	NA	NA	0
** *labA* **	0.01	1.71	0.33	0.5	0.38	0.17
** *oipA* **	3.75	1.71	3.1	2	6	0.67

## Data Availability

Data sharing does not apply to this article as no new data were created or analyzed in this study.

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
