# Peer review of "Determination of the Roles of H. pylori Outer Membrane Virulence Factors and Pyroptosis-Associated NLRP3, ASC, Caspase-1, Gasdermin D, IL-1β, and IL-18 in Ulcer and Gastritis Pathogenesis"

_biology, 2025, doi:10.3390/biology14060634_

Round 1
Reviewer 1 Report
Comments and Suggestions for Authors
In this manuscript, the authors have investigated the association between pyroptosis and the outer membrane virulence factor of H. pylori in patients with gastritis and ulcers. I think this paper will be very informative for the treatment of H. pylori-related diseases. However, I think some of the data in the paper needs to be improved and the discussion needs to be revised. I would like the authors to consider the following correction.
1) (Lane 190-194 and Table 2) In general, patient information for recruited samples should be listed in the “Materials and Methods” section, not in the “Results” section. In addition, do you have any information on their duration of infection, treatment history, medications, etc.? Is the severity of gastric inflammation or ulceration in each patient scored? It would be useful to examine the correlation between the severity of the disease and the expression of various markers.
2) (Table 2) Part of the text “Incidance of Ulcer” is hidden and not visible. In addition, this table is so large that it spans two pages, making it difficult to read. Can't you make the table more compact in height?
3) (Figure legends) The explanation in Figure 1-3 is too simplified and does not give the reader a full understanding. Information such as statistical and analytical methods, n numbers, etc. should be added to the caption of each figure.
4) (Discussion) In the discussion section, many sentences are redundant because they state the results as they are. The authors' thoughts on the results should be more succinctly stated. Furthermore, the authors state “tissue samples taken only from the antrum region may be insufficient to express the general stomach (Lane 494-495)”, so the reasons for limiting the tissue samples to the antrum region of the stomach and papers reporting results from other parts of the stomach should be referenced and discussed. Although this study shows an association between each factor and stomach pathology, it is not a definitive design regarding causality. Therefore, it should be mentioned that specific methods that can elucidate the causal relationship and their analysis are needed.
Author Response
Please see the attachment, and we truly appreciate your careful review and valuable suggestions that strengthened our study.

Reviewer 2 Report
Comments and Suggestions for Authors
In this study, authors investigate the association between pyroptosis and the outer membrane virulence factor of H. pylori in patients with gastritis and ulcers. It’s a very interesting study. Only few questions should be addressed to make it optimized.
- In abstract, conclusions should be grounded in the experimental findings of this study, rather than solely emphasizing their importance and significance.
- Several datasets exhibit disproportionately large standard errors that may undermine statistical validity.,necessitating clarification of error bar conventions (specifically distinguishing between mean±SD and mean±SEM) in the methodology section.
- The observed visual discrepancies in certain datasets that lack statistical significance annotation. It potentially stem from elevated intra-group variability or other methodological limitations requiring systematic investigation?
Author Response
Please see the attachment, and thank you very much for your thoughtful and detailed feedback, it was extremely helpful in revising our work

Round 2
Reviewer 1 Report
Comments and Suggestions for Authors
I checked the corrections in the manuscript. High quality corrections were made to my remarks. I could also feel the authors' strong enthusiasm for this paper. I have no further comments to make. I wish you good luck with this resubmission.